# The Impact of Post-Fire Smoke on Plant Communities: A Global Approach

**DOI:** 10.3390/plants12223835

**Published:** 2023-11-13

**Authors:** Mahboube Zahed, Renata Bączek-Kwinta

**Affiliations:** 1Department of Plant Production, Faculty of Agronomy, University of Agricultural Sciences and Natural Resources in Gorgan, Basij Square, Pardis No. 2, Gorgan 49189-43464, Iran; 2Department of Plant Breeding, Physiology and Seed Science, Faculty of Agriculture and Economics, University of Agriculture in Krakow, ul. Podłuzna 3, 30-239 Kraków, Poland

**Keywords:** plant butenolides, karrikins, seed germination, plant ecology, crop physiology, plant development, swailing, vegetation restoration

## Abstract

Smoke is one of the fire-related cues that can alter vegetation communities’ compositions, by promoting or excluding different plant species. For over 30 years, smoke-derived compounds have been a hot topic in plant and crop physiology. Research in this field was initiated in fire-prone areas in Australia, South Africa and some countries of both Americas, mostly with Mediterranean-type climates. Then, research extended to regions with moderate climates, like Central European countries; this was sometimes determined by the fact that in those regions, extensive prescribed or illegal burning (swailing) occurs. Hence, this review updates information about the effects of smoke compounds on plant kingdoms in different regions. It also focuses on research advances in the field of the physiological effects of smoke chemicals, mostly karrikins, and attempts to gather and summarize the current state of research and opinions on the roles of such compounds in plants’ lives. We finish our review by discussing major research gaps, which include issues such as why plants that occur in non-fire-prone areas respond to smoke chemicals. Have recent climate change and human activities increased the risk of wildfires, and how may these affect local plant communities through physiologically active smoke compounds? Is the response of seeds to smoke and smoke compounds an evolutionarily driven trait that allows plants to adapt to the environment? What can we learn by examining post-fire smoke on a large scale?

## 1. Introduction

Fire has been present on the Earth for millions of years [1,2,3,4]. Fires have been considered an ecological filter that selects plants that possess specific traits [1,2,3]. The responses of plant taxa to fire depend on the fire parameters (duration and frequency) and plant morphology. Regarding post-fire-response traits, resprouting ability and propagule persistence have been identified [4,5]. These characteristics have shifted the focus from studying the effects of fire on individual species to analyzing plant functional traits related to fire in pyrogenic or fire-prone ecosystems. Recently, it has become well known that fire is as an ecological factor that plays a remarkable role in shaping and changing the structures of some plant communities [6,7]. The individual communities undergo the direct impact of fire (heat, mineral accumulation from the burning matter) and/or its indirect impact (smoke). Fire-related cues affect various aspects of plant growth and development, from germination to fruiting [8]. Similar observations for many years served as a pattern for farmers of different regions to make use of prescribed burning to manage ecosystems for agricultural purposes [4].

Seed dormancy is an important trait for many plant species. This is the state in which seeds are unable to germinate, even under proper environmental conditions. Heat shock and the smoke-triggered breaking of seed dormancy, leading to the stimulation of germination, are important complementary factors that affect species with different seed coat permeability [9,10,11]. Smoke produced from the burning of plant biomass needs special attention, as it affects plants through specific chemicals that can alter plant metabolism [12]. For approximately 30 years, intensive efforts have been made to identify the active compounds responsible for smoke-stimulated seed germination. Currently, it is known that at least two groups of fire-related chemicals, karrikins (KARs) and nitriles, present in smoke and its products (aerosol smoke and smoke water) promote germination [13,14]. Karrikins are the most explored group in studies on plant biology and for practical applications in agriculture and horticulture. The stimulatory effects of nitriles such as cyanohydrins, glyceronitrile (2,3-dihydroxypropanenitrile) and mandelonitrile (MAN) on germination have also been determined in several studies [15,16,17]. On the other hand, the discovery of the smoke-related germination inhibitor trimethylbutenolide (TMB) explained why smoke, smoke water and pure smoke compounds affect seeds of the same plant species in different ways [18].

Many studies have provided evidence of smoke-stimulated germination in fire-prone and non-fire-prone ecosystems, in which prescribed or illegal burning (swailing) is periodically carried out [11,16,19,20,21,22,23]. It has to be emphasized that, according to the WHO, the frequency of wildfires is increasing around the globe, as is their severity and duration (https://www.who.int/health-topics/wildfires (accessed on 9 November 2023)) [24]. Investigating the effects of fire products, especially smoke, on plant life allows us to obtain data associated with the responses of different species, and the results can help to determine which taxa benefit from fires, and by which mechanism, and for which taxa fires will be a disadvantage. This serves as a starting point to answer fundamental questions, such as how smoke shapes plant communities in different parts of the globe. Is the smoke response an adaptive trait of plants to regulate seed germination? Is it restricted to regions where the natural burning of vegetation occurs? How do plants perceive smoke signals and respond to them? And what is the general role of smoke in plants? This review aims to discuss the findings associated with smoke compounds, their modes of action and their ecological implications (Figure 1).

## 2. The Impact of Post-Fire Smoke on Plant Communities

### 2.1. Fire-Prone Ecosystems

Examples of plant taxa that show germination stimulation by smoke in regions of the periodic occurrence of fires are indicated in Table 1. Such geographic regions include South Africa and Australia, where the research on the physiological and ecological impacts of smoke-generated volatiles was initiated [25], South America [10], South Asia [26], Africa [8,27], Eurasia [17] and North America [6,28].

#### 2.1.1. Ecosystems with Mediterranean-Type Climates

Mediterranean-type climates (MTCs) are characterized by hot, dry summers and cool, wet winters, and are located between about a 30° and 45° latitude north and south of the Equator, and on the western sides of the continents. Some of these areas are fire-prone, which is why research on the impacts of smoke was initiated there [13,14,15,34,35,36,37]. Studies conducted in MTCs show that the life cycles of many plant species in their fire-prone ecosystems depend on fire. On the other hand, despite comprising only about two percent of the world’s area, these regions are important habitats for one sixth of the world’s vascular plants [4]. Thus, food and species conservation in these regions is considered highly important. In the MTC ecosystem of southeast Australia, the highest frequency of germination is related to obligate seeders (mostly annual species) comprising, among others, the Poaceae species, whose germination is stimulated by smoke [25]. This phenomenon occurs when the species has a rich seed bank and can germinate after the fire event [16]. In these regions, resprouters (plant species that are able to survive fire through the activation of dormant vegetative buds to regrow) can only survive in habitats where the interval between two fires is long enough for regrowth and seed production. However, the trees are suppressed if fires occur frequently in these regions. Therefore, natural selection results in rearrangement of the ecosystem towards grasslands [25].

Annual, perennial and woody species are the dominant plant growth form in the Eastern Mediterranean–Eurasian basin, including Turkey. Çatav et al. [17] reported that annual species play an important role in the post-fire environment characterizing southwestern Turkey. The successful post-fire recruitment of annuals may be due to the enhanced germination of their seeds by smoke-derived chemicals (probably mostly KARs). On the other hand, in this biome, seeds of many woody plants of Fabaceae are unable to germinate due to dormancy, and tend to withstand very high temperatures because their germination is stimulated by heat, not by smoke. In these plants, the presence of various regenerative strategies, including the development of fire-resistant tissues and heat-isolated meristems, resprouting, fire-stimulated flowering, serotiny or fire-stimulated germination, facilitates the post-fire recovery of vegetation [38,39].

#### 2.1.2. Tropical- and Subtropical-Climate Ecosystems

With regard to South America, we should pay attention to the Cerrado biomes in northeastern and southeastern Brazil, dominated by herbaceous and shrubby species. Some species of these regions germinate when exposed to smoke and heat shock during a fire. Smoke appears more effective than heat in South American savannas [10]. However, the high temperatures produced by fire break the seed dormancy of some legumes. On the contrary, temperature fluctuations due to fire in the Cerrado region had little influence on eliminating dormancy in most Fabaceae plants [10]. In fact, physical dormancy protects the seeds of these species against heat, and they emerge after the fire in vegetation gaps due to reduced competition and more availability of light and water [40].

In the West African country of Burkina Faso, the natural vegetation mainly consists of savannas. In this biome, the high concentration of smoke decreases seed germination in some species. Interestingly, the type of smoke can affect natural plant communities differently. For example, aqueous smoke solution did not affect the seed germination of some species, while smoke aerosol was a delaying factor [8,27] (Table 1). The differences in seed response to various smoke forms will be discussed in the Section 3.

Taking Asia into account, the study of native and exotic species in areas affected by anthropogenic fires in another Asian region, Sri Lanka (Patana grasslands, tropical dry forests), shows that plant-derived smoke generally stimulates seed germination, but the sensitivity of plants to smoke can vary. Interestingly, smoke generated from frequent wildfires could potentially increase invasions of weeds and exotic species, which can definitely reshape local plant communities [26]. This means that the increased risk of wildfires, along with the globalization of trade and tourism, may bring new threats.

On the other hand, an experimental study on the role of fire in the germination of seeds of some species native to Florida scrub (North America), with a subtropical savanna and humid climate, revealed smoke-stimulated germination in three of them. Furthermore, almost all species were tolerant to heat shock (consequently to fire). Species’ responses to heat shock can relate to their specific post-fire regeneration strategies [6].

Another experimental study on the role of fire in the germination of montane forest species in the southern portion of North America, Mexico (subtropical area), showed that the response of local plants to fire products strongly depends on the type of dormancy and the fire products, i.e., ash, smoke and heat. Hence, these fire by-products may affect the composition and diversity of species in a post-fire environment [28].

### 2.2. Non-Fire-Prone Ecosystems

Although plant responses to smoke in fire-prone ecosystems are well known as adaptative traits, in non-pyrogenic areas, smoke is not a typical environmental signal for plants. However, some articles have dealt with this, considering various vegetation types and species of different plant taxa [19,23,29,30,31,32,33] (Table 1).

#### 2.2.1. South American Mediterranean Matorral

Mediterranean matorral of Central Chile is devoid of natural fires, but fires of anthropogenic origin have occurred there since the first indigenous settlements (approx. 14,000 BP) and their number has increased since Spanish colonization [19]. That is why Gómez-González et al. [19] evaluated the effects of smoke on the seed germination of 18 woody species native to this region. They revealed that smoke inhibited the germination of eight species of mature matorral communities, but it also stimulated the germination of several pioneer woody species, such as *Acacia caven*, *Trevoa quinquenervia* and *Baccharis vernalis*. Seeds of other species were not affected. An interesting issue was the impact of smoke on the seeds of Asteraceae plants because of differentiated responses (no response, inhibition and stimulation), although in species of the same family, but in fire-prone South African fynbos, the response was almost uniform [19]. The authors proposed phylogenetic constraints as a possible explanation. They also emphasized the possibility of further changes in vegetation composition in the highly human-impacted landscape of the study area.

#### 2.2.2. Tropical Monsoon Ecosystems

An experiment investigating the effect of smoke on the seed germination of 13 species in a monsoon-climate region in China showed that only one species, *Aristolochia debilis*, responded positively to smoke [29]. The lack of germination in the other 12 species may be due to germination-inhibiting compounds in smoke. As smoke is a mixture of many compounds, those species could be considered in further experiments with isolated smoke compounds, in order to investigate their interactions.

#### 2.2.3. Boreal Forests

Many plant species in the boreal forest in North America have physiological dormancy in their seeds. Thus, strategies such as scarification can be applied to alleviate the seed dormancy of native forest species. Fire by-products such as smoke water extract are known to stimulate the germination of seeds from this habitat in North America, but only for seeds treated previously with cold stratification [31]. The reason for this is to enable plants to properly adjust to the environment, because cold stratification occurs naturally from autumn to early spring, mostly in winter, when the temperature is too low to trigger germination, but it alters the seed metabolism gradually towards germination in spring. The same response, namely, smoke sensitivity after cold stratification, was revealed in a herbaceous plant, *Plantago major* (Plantaginaceae), native to Europe and Asia, which was widely introduced through northern and eastern North America [23].

Lamont et al. (2019) [39] emphasized that the response of seeds of a particular species should not be restricted to the geographical region, but is also concerned with the evolutionary history of the species and its ancestors. For example, Poaceae and Brassicaceae, widely dispersed throughout the world, originate from clades that primarily inhabited fire-prone areas. Some representatives of plants from non-fire areas reveal smoke sensitivity originating from their fire-prone ancestors [39]. Such an evolutionary approach has been confirmed in non-fire-prone parts of Europe, where germination rates in Brassicaceae weeds have been reported to be four-fold to five-fold higher than control rates once exposed to smoke [23,30]. This will be further discussed in the Section 6.

#### 2.2.4. European Habitats of Moderate Climate

The first experiments on smoke-stimulated germination among species belonging to European natural flora in non-fire-prone habitats were performed in 2014 in Hungary [30]. Interestingly, smoke water and aerosol smoke greatly increased the seed germination of three Brassicaceae species and *Plantago lanceolata.* The results suggested more successful seedling recruitment for smoke-responsive species with climate-change-driven increases in fire frequency and usefulness in the cultivation of the species or in weed control.

Examples of the impacts of accidental fire and/or smoke in non-fire-prone bio- and agricenoses are gradually increasing. Due to its spectacular, rapid action, it is fire, not smoke, that is considered first. For example, in the Czech Republic, one year after a fire in a species-poor grassland community, a study showed a decrease in the number of plant species. Annual plants (dicots and grasses) were not found, and perennial herbs were prevalent. Herbaceous species eliminated some endemic wind-dispersed species. Hence, the results were opposite to those of pyrogenic areas. Apparently, phytotoxicity in the soil was also responsible for the inhibition of germination and root growth [41].

On the other hand, swailing, used for the regeneration of plant species such as *Calluna vulgaris*, common in the British Isles, led to increased productivity in grasslands [42]. Similarly, the share of rare species possessing underground organs, such as *Epipactis gentiana*, increased in the south of Poland after a fire, but due to the higher content of nutrients in ash [43]. In our opinion, in future research, the impact of smoke should also be analyzed. The problem of smoke-derived compounds emitted during illegal burning in Central Europe was reported by Bączek-Kwinta [23], who revealed that the germination of seeds of *Myosotis arvensis*, a plant native to the moderate climate of Europe and Asia, and abundant in the Central Europe, was strongly suppressed by the application of smoke from burning local meadow plants. On the other hand, seeds of *Trifolium repens*, a globally known and cultivated N_2_-fixer species that is adapted to the moderate climate of Central Europe, responded positively to smoke. This is interesting from an evolutionary point of view, because modern cultivated Fabaceae plants originate from their ancestors from fire-prone environments, but they are usually responsive to heat, not smoke [39].

Investigating the effects of fire on European rangelands, Wójcik and Janicka [44] and Wojcik et al. [43] showed that fire led to a reduction in species richness in a *Molinion* meadow. Habitats of this type are ecologically valuable meadows typical of Central Europe and are protected under the European Natura 2000 network. The studies revealed that fire produced rich and nutritious ash, which was a suitable medium for the growth of rare and extinction-prone species of this habitat. Nonetheless, some studies emphasized that smoke generated by accidental fires can also cause changes in habitat composition in Central Europe, especially in terms of the stimulation of invasive plants like *Solidago gigantea* (giant goldenrod) [23]. For habitats protected by laws due to their natural value, plant invasion is a huge disadvantage. Hence, both the short-term and long-term impacts of fire and its by-products, including smoke, are worth studying.

#### 2.2.5. Asian Habitats

In a study on species from a saline–alkaline grassland in northern China, smoke water significantly increased the germination of salt-tolerant *Setaria viridis* and *Kochia scoparia* seeds [32]. This suggests that smoke can be used as a germination enhancement tool for vegetation restoration in degraded ecosystems.

Another approach was taken to study the impact of smoke, also in northern China, but on non-flowering plants, namely, mosses [33]. As temperate and boreal peatlands in the Northern Hemisphere are important soil carbon stores, and the risk of wildfires increases due to climate change and anthropogenic activity, peatland’s capacity as a carbon sink after a fire depends largely on the successful re-establishment of bryophytes. This is why the effect of smoke water on the spores of 15 moss species was tested. Smoke increased the germination percentage and/or germination speed for 10 species. Interestingly, the germinability of spores naturally buried in peat for 3 to 200 years considerably increased, which indicates the mobilization of spores’ banks. These results definitely indicate that smoke-stimulated spore germination may be common in bryophytes, but also that smoke can be a factor in rearranging the composition of bryophyte species within a peatland community [33].

## 3. General Mechanisms for Perception of Smoke by Plants at the Molecular Level

The active compounds in plant-derived smoke were separated from different plants, showing the diverse nature of smoke based on the plant used to produce smoke. The discovery of these compounds and their implementation into research set the basis for explaining why plants respond to plant-derived smoke and pure smoke compounds differently. Currently, it is well-known that plant-derived smoke contains a variety of stimulants and inhibitors (mostly karrikins and TMB, and trimethylbutenolide, respectively), that can have positive, negative or neutral effects on plant growth, depending on their concentration and exposure period [18,36,45]. Compounds such as cyanide dinitrophenol, azide, fluoride and hydroxylamine have inhibited germination at concentrations approximating those that inhibit metabolic processes [18].

In recent years, various studies have been carried out regarding the response of plants to smoke at the molecular level and the role of smoke in physiological mechanisms in plants. Karrikinolides (KAR1-KAR6) are butenolide compounds that were first discovered as post-fire germination stimulators in plants [14,46]. Present in smoke and its formulations, they affect cell metabolism by interacting with phytohormones such as ABA, GA, IAA and ethylene, which initiates specific signal transduction and gene expression [47]. However, the responses of model and arable plants to karrikin are not always similar and vary in terms of growth, development and photosynthetic characteristics. Nonetheless, karrikin does not result in phytotoxicity [34,35,48].

In general, the detection of KARs by plants is achieved via the expression of a certain gene. In smoke-sensitive plant cells exposed to smoke, first, a receptor protein, KARRIKIN-INSENSITIVE2 (KAI2), detects karrikin, and its structure changes immediately after KAR binding. Then, the protein forms a complex with SMAX2 and SMXL1 proteins, which can degrade the repressor proteins in the KAR signaling pathway. Thus, the activated transcription factor can regulate the expression of certain genes (genes responding to KAR), which eventually leads to specific growth characteristics such as germination, seedling growth [45] and adaptability to abiotic stress [49,50]. A recent study on *Arabidopsis* (a model plant) showed that morphological and growth changes occur in plants if the KAR signaling pathway is interrupted [51]. As was previously mentioned, bryophytes also respond to smoke [33], and time will show whether the molecular mechanism of their response is similar to that of angiosperms.

## 4. The Impact of Smoke and Its Isolated Compounds on Seed Germination and Photosynthesis

Some examples of using smoke formulations or individual smoke-derived compounds are presented in Table 2. KAR1, some nitrogen-containing compounds and syringaldehyde (SLA) were found to stimulate, while TMB inhibited, the seed germination of different species, and MAN had an adverse effect upon the species. Hence, KAR and other smoke-derived compounds can have adverse effects on germination. That is why the impacts of various smoke formulations differ, like pure smoke and smoke-saturated water, as do the impacts of their concentration and the source of the burned plant material (monospecies or a mixture of species) [16,17,18,52,53,54,55,56,57]. The impact of TMB has already been mentioned. Another smoke-derived compound, hydroquinone, is a strong bioactivator that is only active at low concentrations, and at high concentrations, it decreases KAR1 activity and inhibits seed germination [46].

Tavşanoğlu et al. [16] studied both KAR1 and a cyanohydrin analogue, mandelonitrile (MAN), in the seeds of an annual plant, *Chaenorhinum rubrifolium*, that was characterized by strong physiological dormancy. KAR and MAN, used both individually and in combination, stimulated the germination of *Ch. rubrifolium*, and the highest germination rate was achieved via joint treatment with KAR1 and light. Therefore, not only must the smoke-specific molecules be considered, but so must environmental factors characteristic of the local environment. It must be emphasized that the active concentration of KAR differs in smoke produced through the combustion of different plant species [58].

In rice, smoke led to the expression of abscisic acid (ABA)- and gibberellic acid (GA3)-responsive cis-element genes during imbibition. Probably, liquid smoke during imbibition makes it possible for the seed to exploit its reserves to start metabolic activities, which leads to earlier radicle emergence [59]. This is accomplished by stimulating the enzymes required to translocate the seed reserves or through the increased permeability of membranes to growth regulators and phytohormones [23,59].

The mechanism behind increased photosynthetic pigments due to smoke is not yet known, but due to the semi-phytohormonal role of karrikin, it is probable that this bioactive compound stimulates the genes involved in the chlorophyll biosynthesis pathway, or inhibits genes involved in chlorophyll degradation. An increase in chlorophyll concentration was indicated in basil (*Ocimum basilicum* L.) [60]. This definitely enables the plant to intercept solar radiation better, which leads to improved photosynthetic productivity [12]. In future studies, it would be beneficial to study this physiological mechanism with different light intensities, because the ratio of chlorophyll *a* to chlorophyll *b* and carotenoids depends on light affecting the photosynthetic performance.

## 5. Karrikins as Protective Agents against Abiotic Stress

Under various conditions considering abiotic stresses, karrikin increased several abscisic acid signaling pathway genes, including *ABI5*, *ABI3*, *RELATED PROTEIN KINASE2.6*, *(SNF1)-RELATED PROTEIN KINASE2.3* and *(SNF1)-RELATED PROTEIN KINASE2.3*, without an increase in ABA. KAR1 significantly increased organic acid and amino acid content. This shows that karrikins probably decrease abiotic stresses through redox homeostasis. Therefore, karrikins interact directly with ABA-regulating genes to regulate stress adaptability [61]. Improved drought tolerance in herbaceous weeds is associated with the activation of genes responding to karrikin, and the transcription factors of genes related to the increased expression capacity of antioxidants [62]. Karrikins decrease oxidative stresses resulting from drought, salinity and heavy metals by increasing the expression of different enzymes and genes involved in stress mitigation. In fact, karrikins provide a stress tolerance mechanism by controlling the cell antioxidant apparatus and the activities of antioxidants such as superoxide dismutase (SOD) and catalase (CAT) [63].

As was already mentioned, phytohormones such as gibberellin, auxin, abscisic acid and ethylene can regulate plant growth and development. Interestingly, they can either concur with or go against the functions of KARs and strigolactones (SLs), which regulate several developmental processes that adapt the shoot and root architecture to the environment, mostly drought and phosphorus deficit [50]. Such interactions open new possibilities for researchers and potential practical applications like weed control or vegetation restoration, especially in degraded areas (Figure 1).

## 6. Smoke as an Evolutionary Force?

Plants are unable to avoid environmental challenges; hence, they reconstruct metabolic networks and evolve mechanisms to control their structural and functional traits. Then, these stress-induced traits may be used as selection criteria to combat stress, which ensures plant survival under adverse conditions [64,65,66].

Although it is difficult to prove that fire-related traits are adaptive responses, fire and its byproducts (e.g., heat, smoke, char) are now identified as a natural evolutionary force that has shaped and regulated organismal traits [7], provided a fitness benefit following fires and become genetically fixed over time [39]. All Mediterranean-type climate (MTC) habitats, except those of Chile, show a remarkable degree of evolutionary convergence in response to fire in their floras [37,67]. This is reflected in evolutionarily driven traits that allow plants to adapt to local environments. Given that the majority of plant species respond positively to plant-derived smoke in the enhancement of seed germination and plant growth [12], and that the germination of fire-adapted plants is stimulated by compounds possessing butenolide moieties (karrikins) present in smoke [39], it would appear that the most parsimonious adaptive solution is to take advantage of KAR to stimulate post-fire germination. This confirms that if a trait has evolved in response to selection by fire, then the environment of the plant must have been fire-prone before the appearance of that trait [68]. Moreover, since flowering plants more clearly arise in fire-prone environments, this means that the presence of KAR sensitivity among flowering plants can be traced back to their fire-prone ancestors [69,70] (Figure 1).

The KAR sensitivity of non-flowering plants is also worth studying from this standpoint. A new research area regarding the impact of smoke on plants is to estimate the role of fire in the potential regeneration of peatland bryophytes from spores. The promotive effects of smoke on spore germination have demonstrated and proven that smoke water can highly stimulate the germination of ten types of bryophyte spores in another part of the globe, Northeast China [33]. From a scientific standpoint, it would be worth extending this research to other non-seed plants, which are evolutionarily older than angiosperms and gymnosperms, and definitely have an interesting history of coping with fire and its by-products. Such ideas were already raised by Lamont et al., 2019 [39] for seed plants; hence, a shift towards mosses, ferns, etc., would be an interesting step in plant biology and ecology.

## 7. Large-Scale Testing of Smoke from Burnt Vegetation

The increasing risk of fires due to changes in climate and anthropogenic factors means the need to monitor and forecast them. The same applies to smoke, which can be a significant signal of biomass burning, allowing for early fire detection. During a wildland fire, smoke constituents are released and, in some ecosystems, travel over considerable altitudes of several kilometers or are trapped in valleys (Mallia et al., 2020) [71]. Therefore, fire emissions are considered an essential factor for assessing climate and air quality changes. Smoke testing and forecasting are interesting from different viewpoints: vegetation type, instruments and modeling.

### 7.1. Ground- and Aircraft Monitoring and Modeling

Testing post-fire smoke on a large scale is a challenge that requires high-class equipment, state-of-the-art knowledge, safety procedures and close coordination with local authorities and experts in air quality and environmental monitoring. Ground and airborne measurements can be performed. For prescribed burning, experimental plots should be representative of the areas affected by the fire, including the fire’s point of origin and areas downwind. Wind direction and speed have to be monitored. For the testing team, personal protective equipment and safety protocols are necessary. Over the past decade, field-based large fire experiments and related technology, such as Studies of Emissions and Atmospheric Composition, Clouds and Climate Coupling by Regional Surveys (SEAC4RS); WE-CAN campaigns; the Fire and Smoke Model Evaluation Experiment (FASMEE); and the Combustion-Atmospheric Dynamics Research Experiment (RxCADRE), have allowed for the investigation of smoke movement (Garofalo et al., 2019; Ottmar et al., 2016, Prichard et al., 2019) [72,73,74].

Based on the Weather Research and Forecasting Model (WRF) data, smoke dispersion can be modeled using different methods (Mallia et al., 2020 [71,75]). This typically involves assessing air quality by determining the presence of suspended particulate matter (PM), gases (mostly carbon dioxide (CO_2_), carbon oxide (CO) and methane (CH_4_)) and air optical properties (e.g., single-scattering albedo (SSA)) in the aftermath of a fire (Strand et al., 2015) [76]. Accurate estimates of fire and smoke emissions from wildland fires are highly dependent on the amount of fuel contained in an ecosystem, fuel availability (dryness, consumption) and fire behavior, which are predominantly under the control of climate and weather (Sokolik et al., 2019) [77].

To distinguish smoke parameters for different types of vegetation, Strand et al. (2015) [76] performed smoke measurements during swailing of the grass and forest understory in Northwest Florida, United States, using ground- and aerostat, as well as aircraft instrumentation. The results indicated that the particles from the forest burn were less light-absorbing than those from that of grass. On the contrary, CO and CH_4_ emission factors were twofold higher for the forest fire than the grass fire. The reason was the longer duration of smoldering combustion in the forest biomass. The research revealed the evolution of smoke emissions from two co-occurring plant habitats and demonstrated the complexity of emission factors.

Concerning instruments and modeling, a fire–atmosphere model (WRF-SFIRE) with improved forest canopy wind parameterization based on a non-dimensional wind profile was implemented to simulate a prescribed burn within a forested plot in Northwest Florida, United States (Mallia et al., 2020) [71]. The experimental burn plot, with an area of 1.51 km^2^, mostly consisted of trees, allowing the authors to test canopy wind parameterizations. Swailing lasted ca. 3 h. A number of wind anemometers were located throughout the plot. An aircraft equipped with a Wildfire Airborne Sensor Program (WASP) sensor was used to measure the fire’s radiative power, and another measurement platform mounted on a Cessna 337 collected smoke samples, which were analyzed using a Picarro Cavity Ring-Down Spectroscopy (CRDS) gas analyzer. CO_2_, CO, CH_4_ and water vapor were sampled every 2 s before and during swailing. Smoke was detected shortly after the ignition of the experimental plot. Despite some limitations, the coupled methods significantly improved the analysis of sub-canopy winds, fire growth rates and smoke dispersion when evaluated with observations.

Another approach was to use two sources of air quality data to analyze smoke dispersion in central Utah for the 2018 Pole Creek and Bald Mountain Fires. Low-cost, popular calibrated light-scattering PM_2.5_ sensors (AQ&U) from publicly accessible fixed sites were supported by a semicontinuous public transit mobile platform (Transit Authority light-rail Transit Express; TRAX) for PM_2.5_, CO_2_, O_3_, CH_4_ and NO_2_ (Mallia et al., 2020) [75]. Smoke simulations were generated using a coupled fire–atmosphere–chemistry model, WRFSFC, that simulates the interactions between fire behavior, fuel moisture, aerosol feedback and atmospheric conditions at each WRF time interval. The results revealed that this combination of applied methods can be an effective tool to depict the complex spatiotemporal heterogeneity of smoke plumes, also in mountain terrain. This is important, especially when prescribed burning is planned, because in valleys, smoke can be entrapped within shallow layers of air near the ground at night and be carried to unexpected destinations (Achtemeier, 2005; Mallia et al., 2020) [75,78]. Smoke-containing fog is considered a threat to human health and safety, because it triggers respiratory problems and impairs visibility. It can also be considered in further research on vegetation, however.

### 7.2. Satellite Observations

Satellite techniques also allow us to monitor smoke emissions on large scales. In this case, image analysis is used. For example, Sun et al. (2023) [79] proposed fire source identification and positioning based on smoke. The smoke’s spectral characteristics and variation pattern were first studied and analyzed based on a physical correlation model and laboratory experiments. The spectral variation in the vegetation background was measured using the Mahalanobis distance (MD), and MD-based smoke identification and concentration inversion were carried out. Then, extraction of the smoke concentration center and fire source positioning was performed based on the Laplace operator. Then, verification of the method was carried out using Landsat8/OLI imagery datasets of forest burn smoke in Daxing’anling, China (from 2015), and British Columbia, Canada (2017). The results showed that the model can effectively identify smoke pixels, and the proposed MD-based smoke concentration inversion model can quantitatively map the smoke distribution over the studied areas.

However, the optimization of models based on smoke images is important to distinguish smoke columns from clouds (Guede-Fernández et al., 2021) [80]. Developing a new large-scale satellite imagery dataset based on Moderate Resolution Imaging Spectroradiometer (MODIS) data, Ba et al. (2019) [81] continuously optimized their model by training a large number of smoke image datasets (USTC_SmokeRS). The experiment included 6225 image samples from six classes (i.e., cloud, dust, haze, land, seaside and smoke) for more complex land cover types around the world. Using this dataset, the researchers evaluated several learning-based image classification models for smoke detection and proposed SmokeNet, a new convolutional neural network (CNN) model that incorporated spatial- and channel-wise attention into the CNN to enhance feature representation for scene classification.

Lu et al. (2022) [82] tested smoke emissions using a combination of the Visible Infrared Imaging Radiometer Suite (VIIRS), Himawari-8 Advanced Himawari Imager (AHI) data and multiple new-generation satellite observations for peatland, forest, cropland and savanna/grassland in Indonesia during 2015–2020. They used the satellite parameter Fire Radiative Power (FRP), which provides an alternative pathway to the estimation of biomass-burning emissions. FRP is linked to the rates of biomass combustion and fire emissions via a biomass combustion coefficient and smoke coefficient, respectively. The authors suggested that the majority of Indonesian fire emissions are very likely due to land use conversion and drought. Hence, smoke analysis confirms the impact of anthropogenic activity and climate change on fires that disturb vegetation.

## 8. Conclusions

Smoke-derived compounds emitted from massive fires, accidental swailing and prescribed burning can affect crops, trees and natural plant communities differently. In fire-prone ecosystems, smoke permanently shapes the plants’ species composition towards grasslands. In non-fire-prone ecosystems, smoke can lead to their rearrangement, but long-term research on this phenomenon is needed. As the frequency and intensity of massive and local wildfires increase due to climate change and anthropogenic activities, the impact of smoke on local plant communities in the near future could be more likely than before. Unfortunately, smoke can stimulate the germination of seeds of some invasive species, which is extremely dangerous for the conserved natural areas, both pyrogenic and non-pyrogenic.

Laboratory experiments on smoke’s impact on plants, mostly seeds, reveal that the response differs within a species and even among varieties. Smoke formulations linked to different concentrations of stimulators (like KARs) and inhibitors (like TMB) are crucial. Other environmental factors, mostly light, but also temperature, soil composition and specific microbiota, have to be considered, as well. On the other hand, the molecular mechanism of the mode of action of the key smoke compound, karrikin (KAR), is well known, and efforts to study it in different plant taxa will reveal whether it is universal for all plants species that produce smoke-responsive seeds. Interestingly, the evolutionary history of plant taxa, like Poaceae and Brassicaceae, which are widely dispersed and cultivated around the world but whose ancestors originate from fire-prone habitats, explains the KAR sensitivity of their seeds, irrespective of their recent occurrence, namely, fire-prone or non-fire-prone bio-and agricenoses.

Testing post-fire smoke on a large scale comprises ground-, airborne, and satellite methods. The first two comprise chemical and physical air parameters as smoke markers. Satellite-based methods utilize data imagery sets and learning-based image classification models. Their implementation allows us to distinguish the type of burnt vegetation, detect fire sources, map the smoke distribution and reveal fire causes.

## Figures and Tables

**Figure 1 plants-12-03835-f001:**
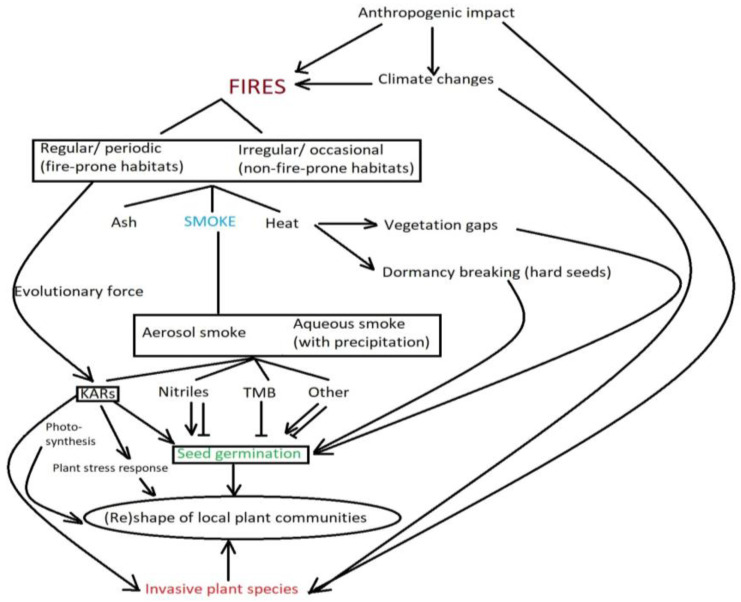
Physiological and ecological impact of smoke on seed germination and local plant communities considering other local and global factors. KARs—karrikins, TMB—trimethylbutenolide.

**Table 1 plants-12-03835-t001:** Examples of plant taxa from different vegetation types that show germination stimulation by smoke formulations. MTC—Mediterranean-type climate.

Region	Vegetation Type	Germination Stimulation with Application of Smoke Formulations		Reference
		Smoke	Smoke Water	
Fire-prone ecosystems				
South Africa and Australia, MTC	Fynbos		*Poaceae* sp.	[25]
Eurasia, Turkey, Mediterranean Basin, MTC	Chaparral		Annual herbaceous species	[17]
South America (Brasil)	Cerrado		*Mimosa somnians*	[10]
			*Cambessedesia hilariana*	
			*Microlicia* sp.	
Africa, Burkina Faso	Sudanian Savanna–Woodland		*Pteleopsis suberosa*	[8]
			*Terminalia avicennioides*	
			*Borreria scabra*	[27]
Asia, Sri Lanka	Savanna–woodland		*Flueggea leucopyrus*	[26]
			*Maesa indica*	
			*Phyllanthus emblica* L.	
			*Chromolaena odorata* L.	
			*Hyptis suaveolens* L.	
North America, Florida	Scrub species		*Chrysopsis highlandsensis*	[6]
			*Eryngium cuneifolium*	
			*Lechea cernua*	
North America, Mexico	Montane forest		*Fuchsia encliandra*	[28]
			*Pinus douglasiana*	
Non-fire-prone ecosystems				
South America, Central Chile, MTC	Matorral	*Acacia caven*		[19]
		*Baccharis vernalis*		
		*Trevoa quinquenervia*		
Asia, South of China	Monsoon climate		*Aristolochia debilis* Siebold and Zucc.	[29]
Central Europe, Hungary	Not specified	*Camelina microcarpa* Andrz. ex DC.		[30]
		*Capsella bursa-pastoris* (L.) Medik		
		*Descurainia sophia* (L.) Webb ex Prant		
		*Plantago lanceolata* L.		
North America	Boreal forest		*Vaccinium myrtilloides* Michx.	[31]
Central Europe, Poland	Not specified/local bio- and agricenoses	*Matricaria chamomilla* L.		[23]
		*Solidago gigantea Aiton* (alien, invasive)		
		*Trifolium repens* L.		
		*Artemisia absinthium* L.		
		*Plantago major* L.		
Asia, Northeast China	Saline–alkaline grasslands		*Setaria viridis* (L.) P.Beauv.	[32]
			*Kochia scoparia* (L.). var. *sieversiana* (Pall.) Ulbr. ex Aschers. et Graebn	
Asia, Northeast China	Northern peatland		*Sphagnum flexuosum* Dozy and Molk	[33]
			*S. subnitens* Russow and Warnst	
			*S.imbricatum* Hornschuch ex. Russow	
			*S. magellanicum* Brid.	
			*S. fuscum* (Schimp.) H.Klinggr	
			*S. squarrosum* Crome	
			*Polytrichum strictum.*	
			*Drepanocladus aduncus* (Hedw.) Warnst.	
			*Physcomitrium sphaericum* Brid	
			*Hypnum callichroum* Hedw.	

**Table 2 plants-12-03835-t002:** Examples of the impacts of various smoke compounds on seed germination and seedling vigor.

Plant Species	Physiologically Active Smoke Compound	Mode of Action	Reference
*Lactuca sativa*	KAR1	Stimulates seed germination	[52]
*Chaenorhinum rubrifolium* Robill. and Castagne ex DC	Mix (aqueous smoke), nitrate	Breakdown of physiological dormancy	[16]
	KAR1, MAN	Stimulates seed germination	
*Ansellia africana* Lindl.	TMB	Reduces the germination rate index and the development rate index	[53]
*Heteropogon contortus* (L.) P.Beauv. ex Roem. and Schult.	Benzaldehyde, cyanide, potassium cyanide	Stimulates seed germination	[54]
*Lactuca sativa* L.	MAN	Inhibits seed germination	[18]
*Nicotiana attenuata* Torr. ex S.Watson	SLA	Stimulates seed germination	[55]
32 plant species belonging to Apiaceae, Asteracea, Boraginaceae, Caryophyllaceae, Cistaceae, Hypericaceae, Lamiaceae, Malvaceae, apaveracea, Poaceae, Polygonaceae and Rosaceae	Glyceronitrile and smoke/butanolide solution	Seed germination and seedling length are enhanced	[17]
*Capsicum annuum* L.	KAR1	Stimulates germination and seedling emergence	[56]
*Daucus carota* L.	KAR1	Positively affects seed germination and plant height	[57]

Abbreviations: KAR1—karrikin 1; MAN—mandelonitrile; SLA—syringaldehyde; TMB—trimethylbutenolide.

## Data Availability

Not applicable.

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
