# Peer review of "The Impact of Post-Fire Smoke on Plant Communities: A Global Approach"

_plants, 2023, doi:10.3390/plants12223835_

Round 1
Reviewer 1 Report
Comments and Suggestions for Authors
This review focuses on the research advances in the field of smoke compounds and summarizes the recent state of opinions on the role of smoke-derived compounds to the germination of plants. The topic is very interesting, and the content falls within the scope of this journal, plants. I have several concerns below which might be of use when revising the ms.
1) At the first glance of the title, I would expect to see a map rather than a simple exhibition of Tables. In this sense, the title is not appropriate (too large) for this version unless the authors supplement more figures or synthesize the global characters of smoke or related compounds in a more straightforward way.
2) A clear scheme illustrating the whole story would work better for readers. And the whole content can follow the scheme logically.
3) English writing should be polished. There are many grammar mistakes or awkward sentences. I only listed a few of them below.
15-18 separate into 2-3 sentences would be better.
26 selects->select
29 basis fire-response traits?
31 So fire is as an ecological factor plays a remarkable?
42 promote germinationv?
45-46 needs to be revised
68-68 needs to be revised
144-147 this sentence is not so logically related to the former one.
166-167 needs to be revised
Comments on the Quality of English Languageneeds to be improved.
Author Response
Dear Editors of Plants,
We have the pleasure to present our manuscript
The impact of post-fire smoke on ecosystems all over the world
in its new form with the new title
The impact of post-fire smoke on plant communities: a global approach.
We did our best to implement the comments and remarks of the Reviewers, to whom we are grateful. The whole text has significantly changed, hence it was impossible to indicate the changes, especially after English Editing by he MDPI Language Service.
We appreciate if you present attached new version of our article to Reviewers.
Sincerely,
Mahboube Zahed & Renata BÄ…czek-Kwinta

Reviewer 2 Report
Comments and Suggestions for Authors
Page 1, line 42. Typo. Extraneous "v" after germination.
Page 2, lines 45-46. Appears to be something missing in this sentence.
Page 2, line 48. "Swailing" is an unfamiliar term that I have never seen used in the fire ecology literature before. Suggest you explain what it means and how it differs from prescribed burning or wildfire or substitute another term here and elsewhere.
Page 2, line 70-71. Something missing - "African and grassland".
Page 2, line 91. Reword "were disable".
Page 4, line 160. Italicize Solidago gigantea.
Page 5, Table 1. Remove period after Lechea.
Page 8, Table 2. Insert space between "Table" and "2" in table heading.
Page 8, line 283. "All Mediterranean type climates".
Page 8, line 290. "This confirms".
Page 8, line 291. Insert comma between "fire" and "then".
Page 9, Reference #9. Reference is not complete; no journal title is given.
Page 11, Reference #45. Fix capitalization of article title.
Comments on the Quality of English LanguageSee above
Author Response

(The authors gave the same response as above.)

Reviewer 3 Report
Comments and Suggestions for Authors
I think you should put in the abstract, the main findings or results of your review. Affect or not the smoke in seed germination or seedling growth? What are the main conclusions?
Perhaps you could help to readers making a diagram to understand how smoke compounds are associated or related with other phytohormones to promote different mechanisms or process in plants.
In the review, I have seen few examples how smoke affect seed germination in plants of tropical ecosystems. Is there not information or data from these ecosystems? Perhaps, you should divide your review in a) temperate ecosystems and b) tropical ecosystems. If you will not include information from tropical ecosystems, you should change the title of this review.
In Table 1, you divided Plant Species in three categories (smoke, smoke water, heat). I haven’t seen a discussion about heat effect in plants. I guess, there is only in the Table 1 but not in the text. Moreover, this is not the subject of discussion in this review.
Some scientific names are accompanied with authors, while others not. Standardize.
Author Response

(The authors gave the same response as above.)

Round 2
Reviewer 1 Report
Comments and Suggestions for Authors
Most of my questions raised last time yet remained. I am disappointed with the current version. I do believe a clear scheme would work better. In this case, I insist that the authors revise their ms, and to be concise is not a good reason.
Author Response
Dear Editors of Plants,
We have improved our manuscript
The impact of post-fire smoke on plant communities: a global approach.
We would like to thank the Reviewer 1 much for his/her opinion. We reconsidered the issue and added a scheme that reflects the content of the manuscript. This helped us to rearrange and improve the text. For example, we have added description of all examples indicated in tables. We also removed one part of the text which did not bring any information of the impact of smoke. Having checked the content of the Table 1 and the cited manuscripts, we remodelled it.
All changes are indicated in blue-green.
Again, we would like to thank you for giving us the chance to improve the manuscript.
We appreciate if you present attached new version of our article to Reviewers.
Sincerely,
Mahboube Zahed & Renata BÄ…czek-Kwinta, 19 October 2023

Round 3
Reviewer 1 Report
Comments and Suggestions for Authors
This manuscript has improved much, and can be accepted.
Author Response
Dear Reviewer,
Thank you very much for your opinion and the possibility to improve our manuscript.
Sincerely,
Mahboube Zahed & Renata BÄ…czek-Kwinta, 7 November 2023